# Enhancing Deep Symbolic Regression via Reasoning Equivalent Expressions

## Abstract

Symbolic regression seeks to uncover physical knowledge from experimental data. Recently a line of work on deep reinforcement learning (DRL) formulated the search for optimal expressions as a sequential decision-making problem. However, training these models is challenging due to the inherent instability of the policy gradient estimator. We observe that many numerically equivalent yet symbolically distinct expressions exist, such as $\log(x_1^2 x_2^3)$ and $2\log(x_1) + 3\log(x_2)$. Building on this, we propose Deep Symbolic Regression via Reasoning Equivalent eXpressions (DSR-REX). The high-level idea is to enhance policy gradient estimation by leveraging both expressions sampled from the DRL and their numerically identical counterparts generated via an expression reasoning module. Our DSR-REX (1) embeds mathematical laws and equalities into the deep model, (2) reduces gradient estimator variance with theoretical justification and (3) encourages RL exploration of different symbolic forms in the search space of all expressions. In our experiments, DSR-REX is evaluated on several challenging scientific datasets, demonstrating superior performance in discovering equations with lower Normalized MSE scores. Additionally, DSR-REX computes gradients with smaller empirical standard deviation, compared to the previous DSR method.

## 1 Introduction

Mathematical modeling of observed phenomena is essential to many scientific and engineering disciplines. Symbolic regression has emerged as a promising approach to automatically discover new physical laws from experimental data (Schmidt & Lipson, 2009; Wang et al., 2019; Udrescu & Tegmark, 2020; Cory-Wright et al., 2024). Recent researchers proposed the use of deep reinforcement learning (DRL) to guide the search for optimal expressions by framing the problem as a sequential decision-making process (Petersen et al., 2021; Landajuela et al., 2022; Jiang et al., 2024).

The main challenge of DRL is the unstable training, which arises primarily from the high variance of the policy gradient estimator (Wu et al., 2018). In literature, the common solution to reduce the variance is to subtract a baseline from the estimator (Weaver & Tao, 2001). Another approach is reward-shaping (Ng et al., 1999), which smooths the reward function in RL by designing an extra potential function. However, this potential is hard to design, since the reward function is sensitive to small modifications in the expression. The rest of the works are discussed in the related work.

We observe that different symbolic formats can represent identical mathematical expressions. For example, $\log(x_1^2 x_2^3)$, $\log(x_1^2) + \log(x_2^3)$, and $2\log(x_1) + 3\log(x_2)$ are a group of numerically equivalent but symbolically distinct expressions. Such a group can be obtained by a symbolic reasoning engine that combinatorially applies mathematical equalities of addition, $\exp, \log$, etc. From the DRL model perspective, this group is obtained by exploring the search space of all expressions using different sequences of step-by-step prediction from the model.

Building on this observation, we introduce Deep Symbolic Regression via Reasoning Equivalent eXpressions (DSR-REX). DSR-REX integrates an existing deep reinforcement learning model with a proposed symbolic reasoning module to accelerate the discovery of governing expressions. By comparing with the existing works, the major advantages of DSR-REX are (1) embedding domain-specific knowledge into the deep model by encoding known mathematical rules, laws, and equalities, (2) achieving variance reduction of the gradient estimator with a theoretical guarantee (in Theo-

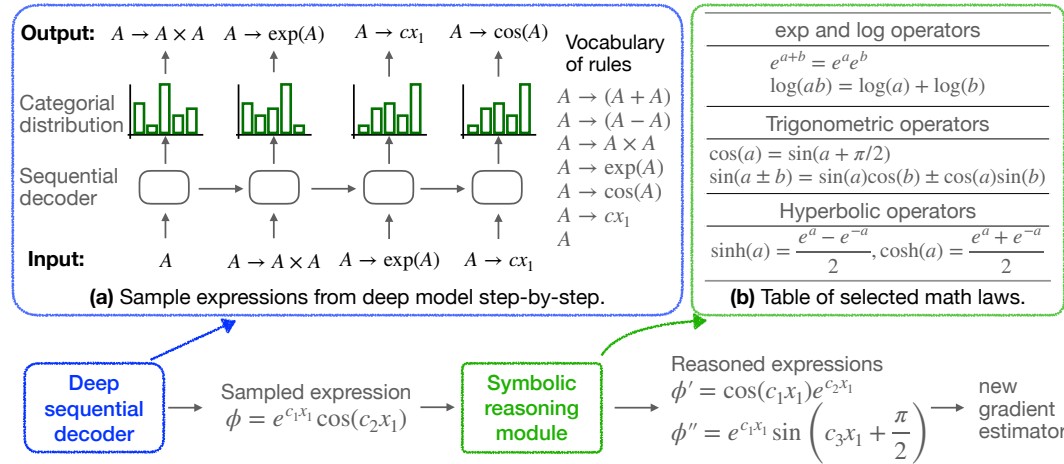

Figure 1: Our DSR-REX integrates symbolic reasoning with reinforcement learning to accelerate symbolic regression, which embeds mathematical equalities into learning, reduces the variance of gradient estimator, and encourages RL exploration. **(a)** In deep reinforcement learning, the deep model predicts an expression by iteratively sampling rules using the probability modeled by the sequential decoder. **(b)** In the proposed reasoning module, mathematical laws are applied to manipulate the input expression to obtain other symbolic-distinct while numerical-equivalent expressions.

rems 1 and 2), and (3) encouraging the exploration of different symbolic forms for DRL model, in the search space of all expressions. In experiments, we demonstrate the advantage of the proposed DSR-REX method over DSR and other baselines by evaluating them on several challenging datasets.

## 2 PRELIMINARIES

**Symbolic Expression.** Let $\mathbf{x} \in \mathbb{R}^n$ be a set of input variables and $\mathbf{c} \in \mathbb{R}^m$ be a set of constants. The expression $\phi$ connects a set of input variables $\mathbf{x}$ and a set of constant coefficients $\mathbf{c}$ by mathematical operators. Variables are allowed to vary and coefficients remain the same. The mathematical operators can be addition, multiplication, etc. For example, $\phi = e^{c_1 x_1} \cos(c_2 x_2)$ is a symbolic expression with one variable $x_1$, two constant $\{c_1, c_2\}$ and three operators $\{\times, \exp, \cos\}$. To cope with the deep reinforcement learning, expression is represented as the traversal sequence of the expression tree (Petersen et al., 2021), the traversal sequence of expression graph (Kahlmeyer et al., 2024), or the sequence of grammar rules (Gec et al., 2022). We adopt the grammar definition in this work, because of it is clear definition and easy integration with the proposed reasoning module.

**Symbolic Regression** aims to discover governing equations from the experimental data. It has been widely applied in diverse scientific domains (Ma et al., 2022; Brunton et al., 2016). Given a dataset $D = \{(\mathbf{x}_i, y_i) | \mathbf{x}_i \in \mathbb{R}^n, y_i \in \mathbb{R}\}_{i=1}^m$ with $m$ samples, symbolic regression searches for the optimal expression $\phi^*$, such that $\phi^*(\mathbf{x}_i, \mathbf{c}) \approx y_i$, where $\mathbf{c}$ denotes the constant coefficients in the expression. From an optimization perspective, $\phi^*$ minimizes the averaged loss on the dataset:

$$\phi^* \leftarrow \arg\min_{\phi \in \Phi} \frac{1}{m} \sum_{i=1}^m \ell(\phi(\mathbf{x}_i, \mathbf{c}), y_i),$$

where $\Phi$ indicates the set of all possible expressions; the loss function $\ell$ measures the difference between the output of the candidate expression $\phi(\mathbf{x}_i, \mathbf{c})$ and the ground truth $y_i$. Since the set of all possible expressions $\Phi$ is exponentially large to the size of input variables and mathematical operators, finding the optimal expression is challenging and is shown to be NP-hard (Virgolin & Pissis, 2022).

**Deep Reinforcement Learning for Symbolic Regression.** A line of recent work proposes the use of deep reinforcement learning (DRL) for searching the governing equations (Abolafia et al., 2018; Petersen et al., 2021; Mundhenk et al., 2021; Landajuela et al., 2022; Jiang et al., 2024). Their

idea is to model the search of different expressions, as a sequential decision-making process using a RL algorithm. The recurrent neural network (RNN) computes the distribution of the next possible symbol given the predicted output. The predicted sequence will then be converted into a valid expression. A high reward is assigned to those predicted equations that fit the dataset well.

Specifically, let $\tau := (\tau_1, \ldots, \tau_k)$ be a sequence composing math operators, variables, and coefficients. The probability $p_\theta(\tau)$ is modeled by the RNN. The reward function $R(\tau)$ computes the goodness-of-fit of the corresponding expression over the dataset $D$. The learning objective is to maximize the expected reward:

$$J(\theta) := \mathbb{E}_{\tau \sim p_\theta} [R(\tau)] \tag{1}$$

$$\nabla_\theta J(\theta) = \mathbb{E}_{\tau \sim p_\theta} [R(\tau) \nabla_\theta \log p_\theta(\tau)], \tag{2}$$

where $\theta$ are the parameters of the neural network and $\nabla_\theta J(\theta)$ is the policy gradient of the expected reward w.r.t. the parameters $\theta$. During training, given $N$ sequences $\{\tau_1, \ldots, \tau_N\}$ sampled from the model with probability $p_\theta(\tau_1), \ldots, p_\theta(\tau_N)$ and the gradient estimator is formulated as follow:

$$\widetilde{\nabla}_\theta J(\theta) = \frac{1}{N} \sum_{k=1}^{N} R(\tau_i) \nabla_\theta \log p_\theta(\tau_i). \tag{3}$$

Afterward, gradient-based optimization is adopted to update the parameters $\theta$ using $\widetilde{\nabla}_\theta J(\theta)$. The estimator $\widetilde{\nabla}_\theta J(\theta)$ is known to have high variance (Sutton & Barto, 1999; Weaver & Tao, 2001). Recent methods have considered several practical strategies to reduce the variance of the estimator and smooth the learning curve. The discussion of these strategies is presented in the related work.

## 3 METHODOLOGY

**Motivation.** We take Figure 1 as an illustrative example to explain a new perspective on symbolic regression. One possible predicted expression is $\phi = e^{c_1 x_1} \cos(c_2 x_1)$, describing the time-evolving behavior of the damped harmonic oscillator. This equation can be symbolically rewritten as $\phi' = \cos(c_2 x_1) e^{c_1 x_1}$ by simply switching the operands of the multiplication operator. Additionally, it can be transformed into $\phi'' = e^{c_1 x_1} \sin(c_2 x_1 + \pi/2)$ using a trigonometric identity, that is $\cos(a) = \sin(a + \pi/2)$. Despite their different symbolic forms, these expressions produce the same numerical output for the same input $\mathbf{x}$, i.e., $\phi(\mathbf{x}) = \phi'(\mathbf{x}) = \phi''(\mathbf{x})$. We refer to such sets of symbolically distinct expressions that yield the same numerical output as *numerically equivalent*. They can be generated by systematically applying mathematical identities or laws. Figure 1(b) shows part of the applicable mathematical laws.

From a reinforcement learning (RL) perspective, generating symbolically distinct expressions corresponds to exploring different subspaces of the expression space, by applying various sequences of grammar rules during decision-making. Denote $\tau, \tau', \tau''$ as three sequences of predicted rules from the RL that can be converted into expression $\phi, \phi', \phi''$ accordingly. We know $\tau \neq \tau' \neq \tau''$. Since the reward function in DRL is based on the error between the output from the predicted expression and the ground truth, all three expressions are assigned the same reward $R(\tau) = R(\tau') = R(\tau'')$. Thus, these expressions are *equivalent* under the RL reward function and are *distinct* under the prediction order of the RL policy.

Our idea is to utilize the sequences sampled from the RL model and additional sequences generated by a symbolic expression reasoning module. These additional sequences capture mathematical equality knowledge and also promote better exploration during RL policy learning. In the space of all possible expressions, the RL model explores those sub-spaces directly sampled according to the policy distribution and additional sub-spaces resulting from different predicted orders of the grammar rules. By integrating this reasoning-driven exploration, our proposed method, DSR-REX, has the potential to discover higher-quality expressions with fewer iterations compared to the Deep Symbolic Regression (DSR) (Petersen et al., 2021).

**Main Procedure.** As depicted at the bottom of Figure 1, DSR-REX consists of three key components: (1) a sequential decoder that samples sequences of grammar rules following its probability distribution step-by-step, (2) a symbolic reasoning module that extracts those equivalent expressions as well as the corresponding sequences of grammar rules, and (3) a parameter update module

that computes the objective and updates the decoder parameters using a gradient-based optimizer. Throughout the training process, the expression with the best goodness-of-fit among all sampled expressions is selected as DSR-REX's final prediction. In the following sections, the problem definition is formulated in section 3.1, and the complete pipeline is presented in section 3.3.

## 3.1 PROBLEM DEFINITION OF DSR-REX

Let $\phi = \text{MAP}(\tau)$ denote the process of converting a sequence $\tau$ into an expression $\phi$. It is internally implemented by converting the sequences into an expression following the grammar definition and then fitting the coefficients of $\phi$ with training data $D = ((\mathbf{x}_1, y_1), \ldots, (\mathbf{x}_m, y_m))$ with a gradient-based optimizer (like BFGS (Fletcher, 2000)). The expressions with fitted coefficients are considered *numerically equivalent* if they can either (1) be derived from each other using symbolic transformations or (2) produce the same output for a large set of random inputs.

We further define *an equivalent group* over a set of sequences if the converted expressions are numerical-equivalent. The probability of each group is defined as the summation of probability for each individual sequence:

$$q_\theta(\phi) := \sum_{\tau \in \Pi} \mathbb{I}\{\text{MAP}(\tau) = \phi\} p_\theta(\tau) \tag{4}$$

where $p_\theta(\tau)$ is the probability of sampling sequence $\tau$ from the sequential decoder, and $\Pi$ is the set of all possible sequences. The indicator function $\mathbb{I}\{\cdot\}$ outputs 1 if sequence $\tau$ can be converted into expression $\phi$; otherwise it outputs 0. In other words, it checks $\tau$ if it belongs to the group indicated by $\phi$. In practice, we do not need to enumerate all sequences in $\Pi$. Equation 4 is defined in this way for the clarity of presentation.

Based on our probability definition in equation 4, the objective together with its gradient becomes:

$$J(\theta) := \mathbb{E}_{\phi \sim q_\theta} [R(\phi)] \tag{5}$$
$$\nabla_\theta J(\theta) = \mathbb{E}_{\phi \sim q_\theta} [R(\phi) \nabla_\theta \log q_\theta(\phi)] \tag{6}$$

For notation simplicity, we assume the reward function $R$ can evaluate the goodness-of-fit for either the expression $\phi$ or the sequence $\tau$ as input. Compared with the classic objective (in equation 1), the main difference is the expectation in equation 5 is over another distribution $q_\theta$. We show in Theorem 1 that our objective is equivalent to the classical formulation. So is the gradient of the objective in the second line. This ensures that DSR-REX and DSR (with no reasoning module) (Petersen et al., 2021) will converge to the same set of optimal parameters.

Since we cannot directly use the probability distribution $q_\theta$ to sample a group of sequences with the same reward. Instead, we only have one sampler that draws sequences from the sequential decoder with probability distribution $p_\theta$. To accommodate this setting, the following estimator is used for the new policy gradient (in equation 6). By draw $N$ sequences from the decoder $\tau_1, \ldots, \tau_N$ with probability $p_\theta(\tau_1), \ldots, p_\theta(\tau_N)$, we compute:

$$\widehat{\nabla}_\theta J(\theta) = \frac{1}{N} \sum_{i=1}^{N} \sum_{\phi \in \Phi} \mathbb{I}\{\text{MAP}(\tau_i) = \phi\} R(\phi) \nabla_\theta \log q_\theta(\phi) \tag{7}$$

where $\sum_{\phi \in \Phi} \mathbb{I}\{\text{MAP}(\tau_i) = \phi\}$ outputs 1 if there exists at least one expression $\phi$ in the space of all expressions $\Phi$ that can be mapped from the sequence $\tau_i$. In practice, equation 7 is not computed by enumerating every expression in $\Phi$ (as indicated by the inner summation). Please see section 3.2 for the detailed steps.

We show in Theorem 2 that this estimator is unbiased and exhibits lower variance than the previous estimator. This implies that the proposed estimator leads to faster convergence and needs fewer iterations required for training than the classic DSR method.

## 3.2 REASONING EQUIVALENT EXPRESSIONS

The implementation of the symbolic reasoning module relies on the expression representation. We first brief the expression representation and present how we generate symbolic variants.

**Expression Representation.** We use a context-free grammar defined by a tuple $\langle V, \Sigma, R, S \rangle$, where $V$ is a set of non-terminal symbols, $\Sigma$ is a set of terminal symbols, $R$ is a set of production rules and $S \in V$ is the start symbol (Todorovski & Dzeroski, 1997; Sun et al., 2023). We use (1) a set of non-terminal symbols representing sub-expressions as $V = \{A\}$. (2) Set of input variables and constants $\{x_1, x_2, \ldots, x_n, \text{const}\}$ as $\Sigma$. (3) Set of rules representing possible mathematical operations such as addition, subtraction, multiplication, and division, as $R$. For example, the addition operation is represented as $A \rightarrow (A + A)$, where the rule replaces the left-hand symbol with the right-hand side. (4) An start symbol $A \in V$. Given a sequence of rules that begin with the start symbol $A$, each rule replaces the first non-terminal symbol $A$ iteratively. The obtained output with only terminal symbols is a valid mathematical expression. Figure 1(a) presents a sequence of grammar rules that corresponds to equation $\phi = e^{c_1 x_1} \cos(c_2 x_1)$.

To generate numerical-equivalent expressions, we use two strategies: (1) directly modifying the sequence of grammar rules through pattern matching with mathematical laws, and (2) manipulating the symbolic form using simplification and transformation rules from libraries like Sympy.

If we are given a sequence of grammar rules, the process begins by converting the sequence into recursive arrays. If a rule contains two non-terminal symbols on the right-hand side, we group the array into two sub-arrays, each representing a sub-expression. The next step involves pattern matching with available mathematical laws, enabling element exchanges within these arrays. After each modification, a copy of the entire array is saved. For example, commutative properties such as $a + b = b + a$ and $a \times b = b \times a$, as well as trigonometric, exponential, and logarithmic identities like $\cos(x - y) = \cos(x)\cos(y) + \sin(x)\sin(y)$, can be applied. A selected list of these mathematical laws is provided in Figure 1(b), while more rules can be found in Appendix Table 2. Finally, the recursive array is flattened back into a sequence, and the sequential decoder is queried for its probability value. Summing these probabilities yields the grouped probability value $q_\theta(\phi)$, as defined in equation 4. Use the example in Figure 1(a), we have:

$$\tau = (A \rightarrow A \times A, A \rightarrow \exp(A), A \rightarrow cx_1, A \rightarrow \cos(A), A \rightarrow cx_1)$$

$$\Rightarrow \text{step 1: } (A \rightarrow A \times A, (A \rightarrow \exp(A), A \rightarrow cx_1), (A \rightarrow \cos(A), A \rightarrow cx_1))$$

$$\Rightarrow \text{step 2: } (A \rightarrow A \times A, \underbrace{(A \rightarrow \cos(A), A \rightarrow cx_1), (A \rightarrow \exp(A), A \rightarrow cx_1)}_{\text{exchange operands of multiplication operator}})$$

$$\Rightarrow \text{step 3: } \tau' = (A \rightarrow A \times A, A \rightarrow \cos(A), A \rightarrow cx_1, A \rightarrow \exp(A), A \rightarrow cx_1)$$

$$\Rightarrow \text{step 4: compute } q_\theta = p_\theta(\tau') + p_\theta(\tau)$$

For modification on the symbolic format of expression, we utilize the Sympy Python package to simplify, factor, or convert the expression into a canonical form. Each of the available operations will return one symbolic variant. Sympy applies a broader set of pattern-matching rules to transform the expressions. As an additional step, each new expression is converted back into a sequence of grammar rules based on context-free grammar.

It is important to note that the number of equivalent expressions can grow exponentially through various augmentations. For instance, given an expression $\phi$, one can generate infinitely many distinct expressions by introducing and canceling a sub-expression $\phi_e$, such as $\phi + \phi_e - \phi_e$ or $\phi \times \phi_e / \phi_e$. We do not consider the above cases in implementation. Still, we can generate $2^n$ distinct expressions for $x_1 + \ldots + x_n$ by randomly reordering the operands of the summation. To prevent the group size from becoming too large, we introduce a hyperparameter (`max-group-size`) to limit the number of expressions in each group.

### 3.3 THE LEARNING PIPELINE OF DSR-REX

Expression generation begins with the decoder sampling a sequence of grammar rules in an autoregressive manner. This decoder can be implemented using various architectures such as RNNs (Salehinejad et al., 2017), GRUs (Chung et al., 2014), LSTMs (Greff et al., 2016), or Decoder-only Transformer (Vaswani et al., 2017). The input and output vocabularies consist of grammar rules that encode input variables, coefficients, and mathematical operators. Figure 1(a) illustrates an example of output vocabulary.

The model predicts the categorical probability of the next token at each time step, conditioned on the previously generated tokens as the input context. At the $t$-th step, the decoder (denoted as

---

**Algorithm 1** Deep Symbolic Regression via Reasoning over Equivalent Expressions.

---

**Input:** #input variables $n$; Mathematical operators $O_p$; Training data $D$; Sequential decoder.
**Output:** The best-predicted expression $\phi$.
 1: initialize the set of best predicted expressions $\mathcal{Q} \leftarrow \emptyset$.
 2: construct grammar rules from $O_p$ and variables $\{x_1, \ldots, x_n\}$.
 3: set input and output vocabulary with the grammar rules.
 4: **for** $k \leftarrow 1$ *to* #epochs **do**
 5:     sample a batch of sequences $\{s_1, \ldots, s_N\}$ from the sequential decoder.
 6:     construct expressions $\phi_i$ from grammar rules $\tau_i$, for $i = 1$ to $N$.
 7:     fitted coefficients $c_i \leftarrow \text{BFGS}(\phi_i, D)$, for $i = 1$ to $N$.
 8:     saving tuple $\langle c_i, \phi_i \rangle$ into $\mathcal{Q}$, for $i = 1$ to $N$.
 9:     reasoning extra sequences following section 3.2.
10:     compute the estimated policy gradient $\widehat{\nabla}_\theta J(\theta)$ (in equation 6).
11:     update parameters of decoder $\theta^{k+1} \leftarrow \theta^k + \alpha \widehat{\nabla}_\theta J(\theta)$.
12: **return** the best-predicted equation in $\mathcal{Q}$.

---

SequentialDecoder) takes the output from the previous step, $\tau_t$, and the hidden state $\mathbf{h}_t$. It then computes the categorical probability distribution over the vocabulary using the softmax function:

$$\mathbf{z}_t = \text{SequentialDecoder}(\tau_t, \mathbf{h}_t)$$
$$p_\theta(\tau_{t+1} | \tau_1, \tau_2, \ldots, \tau_t) = \text{softmax}(\mathbf{z}_t W_o + b_o)$$

where $W_o \in \mathbb{R}^{d \times |V|}$ is the output weight matrix, $b_o \in \mathbb{R}^{|V|}$ is the bias term, and $|V|$ is the size of the output vocabulary. The next token $\tau_{t+1}$ is sampled from the categorical distribution $\tau_{t+1} \sim p(\tau_{t+1} | \tau_1, \tau_2, \ldots, \tau_t)$. The output from each step is recursively used as the input for the subsequent step, progressively generating the entire sequence. After $L$ steps, the full sequence $\tau = (\tau_1, \ldots, \tau_L)$ is generated, with its probability given by $p_\theta(\tau) = \prod_{t=1}^{L-1} p_\theta(\tau_{t+1} | \tau_1, \ldots, \tau_t)$. Since $\tau_1$ is the fixed start symbol, $p_\theta(\tau_1) = 1$ is omitted here.

The function $\text{MAP}(\tau)$ is then called to convert the sequence into an expression. If the sequence ends before a complete expression is formed, grammar rules representing variables or constants are randomly appended. Conversely, if a valid expression is produced before the sequence is fully consumed, the remaining grammar rules are discarded, and the expression is returned. The probability value $p_\theta(\tau)$ is updated accordingly whenever grammar rules are added or removed.

For each sequence sampled from the decoder, we (1) obtain all additional expressions using possible mathematical rules, (2) reconstruct the corresponding sequence $\tau'$ based on the expression grammar definition, and query the sequential decoder for its probability value $p_\theta(\tau')$ for each additional expression $\phi'$, and (3) compute $q_\theta$ using Equation 4 for each group of probability values.

The objective of DSR-REX is to maximize the probability of sampling expressions that fit the data well. This is achieved through a reinforcement learning objective, where the reward function computes the goodness-of-fit of the sampled expression to the data. The new gradient estimator $\widehat{\nabla}_\theta J(\theta)$ is then used to compute the gradient with respect to the neural network parameters, as shown in Equation 7. At the $k$-th iteration, the parameters are updated using gradient-based optimization. The overall pipeline is summarized in Algorithm 1.

### 3.4 THEORETICAL INSIGHT ON THE ADVANTAGE OF DSR-REX

Theorem 1 establishes that the objective of DSR-REX is equivalent to that of classic Deep Symbolic Regression (DSR), and similarly, their gradients are identical. This implies that DSR-REX and DSR will converge to the same set of optimal parameters. Consequently, after the convergence of the DSR-REX and DSR, they will sample expressions with identical rewards with a high probability.

**Theorem 1. (1)** The expectation of reward over probability distribution $p_\theta(\tau)$ equals the expectation over probability distribution $q_\theta(\phi)$, that is:

$$\mathbb{E}_{\tau \sim p_\theta}[R(\tau)] = \mathbb{E}_{\phi \sim q_\theta}[R(\phi)].$$

**(2)** The expectation of *policy gradient* over probability distribution $p_\theta(\tau)$ equals the expectation over probability distribution $q_\theta(\phi)$, that is:

$$\nabla_\theta J(\theta) = \mathbb{E}_{\tau \sim p_\theta}[R(\tau)\nabla_\theta \log p_\theta(\tau)] = \mathbb{E}_{\phi \sim q_\theta}[R(\phi)\nabla_\theta \log q_\theta(\phi)].$$

*Sketch of the proof.* The result can be obtained by expanding the terms according to the proposed problem definition in section 3.1. The full proof is provided in Appendix B. $\square$

We also demonstrate that DSR-REX provides an unbiased gradient estimator and reduces the variance of the gradient estimate, as shown in Theorem 2).

**Theorem 2.** Using $N$ seqeunces $\tau_1, \ldots, \tau_N$ drawn according to the probability distribution $p_\theta$. **(1)** Unbiased estimator. The expectation of $\widehat{\nabla}_\theta J(\theta)$ over distribution $p_\theta(\tau)$ equals to $\nabla_\theta J(\theta)$, that is

$$\nabla_\theta J(\theta) = \mathbb{E}_{\tau \sim p_\theta}\left[\widehat{\nabla}_\theta J(\theta)\right] = \mathbb{E}_{\phi \sim q_\theta}[R(\phi)\nabla_\theta \log q_\theta(\phi)]$$

**(2)** Variance reduction. The variance of the proposed estimator $\widehat{\nabla}_\theta J(\phi)$ is smaller than the original estimator $\widetilde{\nabla}_\theta J(\theta)$, that is

$$\mathbb{V}\mathrm{ar}_{\phi \sim q_\theta}\left[\widehat{\nabla}_\theta J(\phi)\right] \leq \mathbb{V}\mathrm{ar}_{\tau \sim p_\theta}[\widetilde{\nabla}_\theta J(\theta)]$$

*Sketch of the proof.* For unbiasedness, we show the two estimators in Equations 3 and 7 equals to each other based on Theorem 1. In terms of variance reduction, the key insight is (1) the number of samples with grouping is larger. (2) Since the reward is the same in the group, the variance within the group is smaller. The full proof is provided in Appendix C. $\square$

## 4 RELATED WORK

**Reinforcement Learning for Scientific Discovery.** Recent advancements in artificial intelligence, particularly in deep reinforcement learning (RL), have demonstrated its potential for automating discoveries across various scientific fields (Kirkpatrick et al., 2021; Jumper et al., 2021; Wang et al., 2023). Early work in this area focused on learning symbolic representations of scientific concepts (Bradley et al., 2001; Bridewell et al., 2008). In domains such as materials discovery and chemical engineering, RL agents have been applied to propose novel materials with desirable properties (Beeler et al., 2024; Popova et al., 2018).

**Variance-Reduced Policy Gradient.** Several techniques have been introduced to reduce the variance of policy gradient estimates, a common challenge in reinforcement learning. One widely used approach is the control variate method, where a baseline is subtracted from the reward to stabilize the gradient (Weaver & Tao, 2001). Recent developments, such as Trust Region Policy Optimization (TRPO) (Schulman et al., 2015; Zhang et al., 2021) and Proximal Policy Optimization (PPO) (Schulman et al., 2017), leverage second-order information to enhance training stability. Other approaches, such as reward reshaping (Zheng et al., 2018), modify rewards for specific state-action pairs. Inspired by stochastic variance-reduced gradient methods (Johnson & Zhang, 2013; Deng et al., 2021), Papini et al. (2018) proposed a variance-reduction technique tailored for policy gradients. Unlike these methods, our proposed DSR-REX is the first to reduce variance through symbolic reasoning over expressions, providing a novel contribution to this field.

**Symbolic Regression with Domain Knowledge.** Recent efforts have explored incorporating physical and domain-specific knowledge into the symbolic discovery process. AI-Feynman (Udrescu & Tegmark, 2020; Udrescu et al., 2020; Keren et al., 2023; Cornelio et al., 2023) constrained the search space to expressions that exhibit compositionality, additivity, and generalized symmetry. Similarly, Tenachi et al. (2023) encoded physical unit constraints into equation sampling to eliminate physically impossible solutions. Other works, such as (Bendinelli et al., 2023; Kamienny, 2023), further constrained the search space by integrating user-specified hypotheses and prior knowledge, offering a more guided approach to symbolic regression.

**Thinking Fast and Slow.** The interplay between fast and slow cognitive processes is a key feature of human intelligence (Kahneman, 2011; Anthony et al., 2017; Booch et al., 2021). We argue that

rather than relying solely on the brute-force approach of learning from big data and extensive computation (fast thinking), incorporating careful meta-reasoning to guide the discovery of ground-truth equations (slow thinking) can lead to more efficient and effective outcomes.

# 5 EXPERIMENTS

## 5.1 EXPERIMENT SETTINGS

We consider the Trigonometric dataset (Jiang & Xue, 2023), where each group contains 10 randomly sampled expressions. Also, we select 10 challenging equations from the Feynman dataset (Udrescu et al., 2020). In terms of baselines, we consider a series of methods based on the deep reinforcement learning model: Priority queue training (PQT) (Abolafia et al., 2018), Vanilla Policy Gradient (VPG) (Williams, 1992), Deep Symbolic Regression (DSR) (Petersen et al., 2021), and Neural-Guided Genetic Programming Population Seeding (GPMeld) (Mundhenk et al., 2021).

**Evaluation Metrics.** The goodness-of-fit indicates how well the learning algorithms perform in discovering underlying expressions. We use the normalized mean-squared error (NMSE) of the best-predicted expression by each algorithm, on a separately-generated testing dataset. Given a testing dataset $D_{\text{test}} = \{(\mathbf{x}_i, y_i)\}_{i=1}^n$ generated from the ground-truth expression, we measure the goodness-of-fit of a predicted expression $\phi$, by evaluating the normalized-mean-squared-error (NMSE):

$$\text{NMSE}(\phi) = \frac{1}{n\sigma_y^2} \sum_{i=1}^n (y_i - \phi(\mathbf{x}_i))^2 \tag{8}$$

The empirical variance $\sigma_y = \sqrt{\frac{1}{n} \sum_{i=1}^n \left(y_i - \frac{1}{n} \sum_{i=1}^n y_i\right)^2}$. We use the NMSE as the main criterion for comparison in the experiments and present the results on the remaining metrics in the case studies. The main reason is that the NMSE is less impacted by the output range.

## 5.2 EXPERIMENTAL ANALYSIS

**Regression on Algebraic Equations.** In Figure 2(a), we present the top-ranked equations discovered by the proposed DSR-REX compared to baseline methods, evaluated using the NMSE metric. The quantiles $(25\%, 50\%, 75\%)$ of NMSE demonstrate that DSR-REX consistently identifies better expressions than the baselines after multiple learning iterations. This improvement is primarily due to the generated symbolic variants, which guide the model to strategically explore a broader search space of expressions.

We also compare the empirical mean and standard deviation of the loss for DSR-REX and DSR in Figure 2. The computation details for each estimator are provided in Appendix D.2. Our results

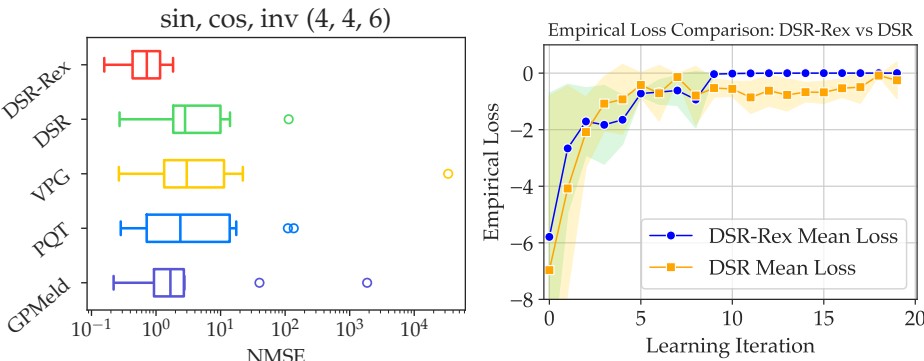

Figure 2: **(Left)** Quantiles $(25\%, 50\%, 75\%)$ of NMSE values for discovered equations across all methods. DSR-REX outperforms baselines due to the generated symbolic variants, which encourage more strategic exploration of the expression search space. **(Right)** Empirical mean and standard deviation of the loss for DSR-REX and DSR, with DSR-REX showing a lower empirical deviation.

show that DSR-REX achieves a smaller empirical deviation than DSR. This reduced variance can be attributed to the group of expressions obtained through symbolic reasoning, which allows us to compute a grouped probability value more efficiently.

**Time Benchmark of DSR-REX.** Figure 3 presents a time benchmark of the four key steps in DSR-REX: (1) sampling sequences, (2) fitting expression coefficients to data, (3) reasoning over additional expressions, and (4) computing the loss, gradients, and updating neural network parameters. We benchmarked three neural network architectures: three-layer LSTM (a), three-layer GRU (b), and six multi-head self-attention layers (c). Our results show that symbolic reasoning is faster than both coefficient fitting and parameter updates. The experimental configuration details are provided in Appendix D.3.

This efficiency is largely attributed to the fact that symbolic manipulations based on mathematical laws do not require refitting coefficients for each modified expression, significantly reducing computational overhead.

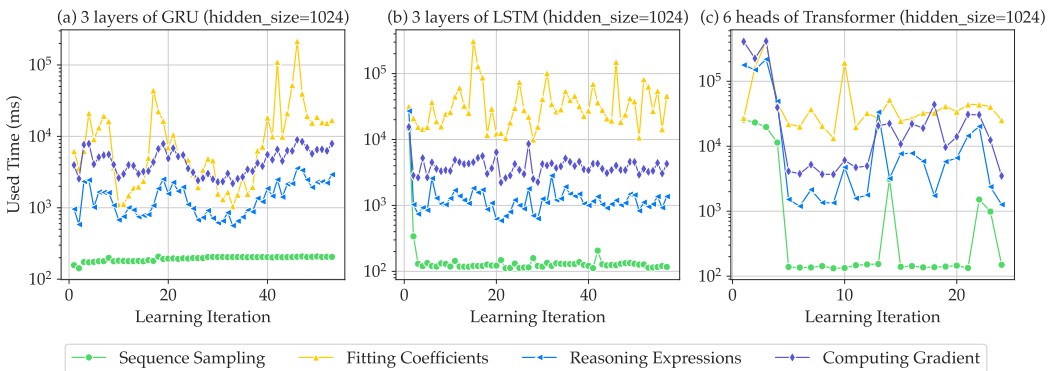

Figure 3: Empirical time benchmark for the four main steps of DSR-REX across different neural architectures: (a) three-layer LSTM, (b) three-layer GRU, and (c) six multi-head self-attention layers. Symbolic reasoning proves to be faster than fitting coefficients and updating parameters.

**Case Studies in DSR-REX.** In addition to the example equations shown in Figure 1, we provide further case studies from the Feynman dataset (Udrescu et al., 2020) to demonstrate the symbolic variants generated by DSR-REX. Table 1 illustrates several symbolic transformations, obtained through symbolic derivation steps, that retain numerical equivalence. These case studies highlight DSR-REX's capability to discover and reason over symbolic variants of complex physical equations.

| Equation | Symbolic variants obtained by DSR-REX |
|---|---|
| $I = I_0 \frac{\sin^2(n\theta/2)}{\sin^2(\theta/2)}$ | $I = I_0 \frac{1-\cos(n\theta)}{1-\cos(\theta)}$ |
| $\kappa = 1 + \frac{N\alpha}{1-N\alpha/3}$ | $\kappa = 1 + \frac{3}{3/N\alpha - 1}$ |
| $Q = nkT \ln(V_2/V_1)$ | $Q = nkT(\ln(V_2) - \ln(V_1))$ |
| $x_1 = \frac{x-ut}{\sqrt{1-u^2/c^2}}$ | $x_1 = \frac{c(x-ut)}{\sqrt{c^2-u^2}}$ |
| $E = \frac{p}{4\pi\epsilon}\frac{3\cos\theta\sin\theta}{r^3}$ | $E = \frac{3p}{8\pi\epsilon r^3}\sin(2\theta)$ |
| $M = N\mu \tanh(\mu B/kT)$ | $M = N\mu \frac{e^{\frac{2\mu B}{kT}}-1}{e^{\frac{2\mu B}{kT}}+1}$ |
| $I_{12} = I_1 + I_2 + 2\sqrt{I_1 I_2}\cos(\delta)$ | $I_{12} = \left(\sqrt{I_1} + \sqrt{I_2}e^{i\delta}\right)^2$ |
| $\phi = \frac{N}{\exp(\mu B/kT)+\exp(-\mu B/kT)}$ | $\phi = \frac{N}{2\cosh\left(\frac{\mu B}{kT}\right)}$ |
| $x = K(\cos(\omega t) + \epsilon\cos^2(\omega t))$ | $x = K\cos(\omega t)\left(1 + \epsilon\cos(\omega t)\right)$ |

Table 1: Case studies showcasing the reasoning module of DSR-REX through symbolic variants obtained from the Feynman dataset (Udrescu et al., 2020).

## 6 CONCLUSION

In this paper, we presented Deep Symbolic Regression via Reasoning Equivalent eXpressions (DSR-REX), a novel approach that enhances deep reinforcement learning with symbolic reasoning. DSR-REX effectively leverages mathematically equivalent expressions to stabilize the policy gradient estimator, reducing its variance and encouraging exploration across the search space. Our theoretical justification and empirical results demonstrate that DSR-REX not only improves gradient estimation but also outperforms existing DRL-based methods in discovering governing equations from real-world scientific data.

In terms of future work, we plan to include laws for vector-field operators, like `div`, `curl`, and `Laplacian` operators. Another possible future direction is to give theoretical convergence analysis for DSR-REX.

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
