# OpenReview forum: "Enhancing Deep Symbolic Regression via Reasoning Equivalent Expressions"
_ICLR.cc/2025/Conference — ICLR 2025 Conference Withdrawn Submission_

### Official Review · Reviewer_ej1f · 2024-10-28

**Soundness:** 2
**Presentation:** 3
**Contribution:** 2
**Rating:** 3
**Confidence:** 4

**Summary:**

The performance comparison between DSR-REX and previous models like DSR and NGGP highlights its superiority. The complexity of the case study equations effectively showcases the model’s symbolic regression capabilities. Additionally, the paper provides clear details of the algorithm and experimental processes.

Weaknesses:
Questions:

**Strengths:**

The performance comparison between DSR-REX and previous models like DSR and NGGP highlights its superiority. The complexity of the case study equations effectively showcases the model’s symbolic regression capabilities. Additionally, the paper provides clear details of the algorithm and experimental processes.

**Weaknesses:**

The paper lacks comparisons with other tasks beyond DSR, such as SPL[1], TPSR[2], and uDSR[3], across different benchmarks like SRbench[4]. It also does not discuss how this method could be applied to these models.

[1]Sun F, Liu Y, Wang J X, et al. Symbolic physics learner: Discovering governing equations via monte carlo tree search[J]. arXiv preprint arXiv:2205.13134, 2022.

[2]Shojaee P, Meidani K, Barati Farimani A, et al. Transformer-based planning for symbolic regression[J]. Advances in Neural Information Processing Systems, 2023, 36: 45907-45919.

[3]Landajuela M, Lee C S, Yang J, et al. A unified framework for deep symbolic regression[J]. Advances in Neural Information Processing Systems, 2022, 35: 33985-33998.

[4]La Cava W, Orzechowski P, Burlacu B, et al. Contemporary symbolic regression methods and their relative performance[J]. arXiv preprint arXiv:2107.14351, 2021.

**Questions:**

1. Could you provide DSR-REX’s results on SRBench[4] or SRSD-Feynman[5] to assess the model's stability under varying noise levels and complexities with scientific implications?

2. What level of improvement might your method bring if applied to other models like SPL[1], TPSR[2], and uDSR[3] for training?

3. Could you share the recovery rate for each expression in Chapter 5.2, Experimental Analysis?

4. Could you include an ablation study on parameters in Appendix Section D?

5. Could you compare DSR-REX with models like SPL, TPSR, and uDSR [1, 2, 3] in the experiments in Chapter 5？

[1]Sun F, Liu Y, Wang J X, et al. Symbolic physics learner: Discovering governing equations via monte carlo tree search[J]. arXiv preprint arXiv:2205.13134, 2022.

[2]Shojaee P, Meidani K, Barati Farimani A, et al. Transformer-based planning for symbolic regression[J]. Advances in Neural Information Processing Systems, 2023, 36: 45907-45919.

[3]Landajuela M, Lee C S, Yang J, et al. A unified framework for deep symbolic regression[J]. Advances in Neural Information Processing Systems, 2022, 35: 33985-33998.

[4]La Cava W, Orzechowski P, Burlacu B, et al. Contemporary symbolic regression methods and their relative performance[J]. arXiv preprint arXiv:2107.14351, 2021.

[5]Matsubara Y, Chiba N, Igarashi R, et al. Rethinking symbolic regression datasets and benchmarks for scientific discovery[J]. arXiv preprint arXiv:2206.10540, 2022.

---

> ### Author Response · Authors · 2024-11-26
> **Thank you for your valuable feedback.**
>
> ### 1. Limited Evaluation Dataset and Baselines
> Thank you for your valuable feedback on the experimental evaluation. We greatly appreciate your suggestions and will incorporate the recommended datasets and recent baselines in a future revision to enhance the comprehensiveness of our evaluation. Additionally, we will conduct further experiments on metrics such as recovery rate to provide a more robust analysis.
>
> We also believe that the proposed module can be empirically applied to and compared against other existing methods. We will include these comparisons in our future revisions to strengthen the empirical evaluation.
>
> ---
>
> ### 2. Ablation Study on Hyperparameters in the Proposed Module
> Thank you for highlighting the need for an ablation study. We acknowledge the importance of evaluating the impact of hyperparameters, such as group size, on the estimated policy gradient value. We will perform this analysis and include the results in a future revision to provide deeper insights into the behavior of the proposed module.

---

> > ### Comment · Reviewer_ej1f · 2024-11-26
> >
> > Thank you for your response. As the experiment is not yet complete, I will not adjust my evaluation score at this time. However, I look forward to your future updates, including detailed information on evaluations across multiple datasets and baselines, as well as an ablation study on hyperparameters.

---

### Official Review · Reviewer_Pc43 · 2024-11-02

**Soundness:** 2
**Presentation:** 1
**Contribution:** 2
**Rating:** 3
**Confidence:** 3

**Summary:**

The paper proposes DSR-REX, which improves the performance of the algorithm by embedding mathematical laws and equalities into deep models. Moreover, the variance of the gradient estimator is theoretically guaranteed to decrease. Finally, in various experimental tests, DSR-REX shows good performance.

**Strengths:**

##### Strengths

1. In this paper, DSR-REX has achieved good results in comparison with other baselines.
2. Achieving variance reduction of the gradient estimator with a theoretical guarantee.

**Weaknesses:**

##### Weaknesses

1. I think the chapter arrangement of this article is unreasonable. For example, the related work is actually behind the method, which makes the article very messy.  I spent an hour not understanding what the author was doing.  I think The **Related work** can be put behind the **Introduction**. The  **Motivation ** part of  **METHODOLOGY ** can be appropriately deleted and put into the  **introduction ** part...
2. Many related works are not mentioned.
**Reinforcement Learning for Scientific Discovery.** such as TPSR(Discovering mathematical formulas from data via gpt-guided Monte Carlo tree search), SR-GPT(Discovering mathematical formulas from data via gpt-guided monte carlo tree search),  RSRM(Reinforcement Symbolic Regression Machine)...

**Symbolic Regression with Domain Knowledge:** NSRwH(Controllable neural symbolic
regression), MLLM-SR(MLLM-SR: Conversational Symbolic Regression base Multi-Modal Large Language Models
),  LLM-
SR(LLM-SR: Scientific Equation Discovery via Programming with Large Language Models)...
3. This article only mentioned the symbolic regression method using reinforcement learning, but symbolic regression is not the only one, other methods should appear in the comparison method, e.g. SNIP,(https://doi.org/10.48550/arXiv.2310.02227) MMSR(https://doi.org/10.1016/j.inffus.2024.102681), DSO(NGGP)(https://doi.org/10.48550/arXiv.2111.00053), TPSR(Transformer-based Planning for Symbolic Regression), and so on
4. The author should test your algorithm on the SRBench dataset.

**Questions:**

##### Questions

1. The third innovation point of the paper, 'Encours-ages RL exploration of different symbolic forms in the search space of all expres- sions' Is to make the probability of model sampling more random? Like adding entropy loss?
2. Article line 151, additional sequences generated by a symbolic expression reasoning module. How does symbolic expression reasoning module generate additional sequences and what is their role?
3. In Figure 1, the **Reasoned expressions** can improve the performance of the algorithm. Please analyze the reasons for the improvement in the performance of the algorithm more carefully in the article.
4. Although your idea is good, I think it is inappropriate for the words "high-level idea" to appear in an academic paper.

---

> ### Author Response · Authors · 2024-11-26
> **Thank you for your valuable feedback!**
>
> ### 1. Paper Presentation and Writing
> Thank you for your feedback regarding the paper's organization. Our intention was to first present the novel methodology and then contrast it with prior work to highlight our contributions. In the revised version, we will reorganize the paper to improve clarity and flow.
>
> ### 2. Missing Related Work
> We appreciate your comments regarding the related work section. We acknowledge that a more comprehensive review of the literature would provide a better context for our contributions. In future revisions, we will include discussions on the additional models you mentioned.
>
> ### 3. Clarification on the Third Innovation Point
> We apologize for any confusion caused by the description of the third innovation point. Our intention was to convey that the extra equations assist the Deep RL model in exploration. This is conceptually similar to the idea of an "empty loss," which prevents the model from becoming overly confident in specific expressions. We will carefully revise this section to ensure the message is rigorous and clear.
>
> ### 4. Clarification on Line 151
> The position of the proposed module is described in Line 159, while the mechanism behind it is explained in Line 240. We will ensure this relationship is explicitly referenced to avoid any confusion in the revised manuscript.
>
> ### 5. Analysis of the Improvement Brought by the Proposed Module
> Thank you for your valuable feedback. The theoretical benefits of the proposed module are outlined in Theorems 1 and 2. The key idea is that the module generates additional equations, reducing the variance of sample estimates and thereby improving the quality of estimation. This, in turn, enhances the model's stability. We will expand the analysis in the revised manuscript to provide further clarity and detail.
>
> ### 6. Inappropriate Use of "High-Level Idea"
> Thank you for pointing out the inappropriate use of terminology. We will replace "high-level idea" with more precise and formal phrasing to maintain academic rigor throughout the paper.

---

> > ### Comment · Reviewer_Pc43 · 2024-12-01
> >
> > Thank you very much for the clarification of the author. Considering your clarification of all the reviewers, I decided to maintain my current score. I wish you every success in your work.

---

### Official Review · Reviewer_p6DY · 2024-11-03

**Soundness:** 2
**Presentation:** 2
**Contribution:** 3
**Rating:** 5
**Confidence:** 2

**Summary:**

This paper proposes DSR-Rex, which adds a mathematical equivalence reasoning module to deep symbolic regression to improve the efficiency and stability in the training process (variance reduction). Based on a re-expression of the objective after grouping mathematically equivalent but symbolically different equations, the algorithm uses standard encoding/decoding modules of DSR plus a novel reasoning module that enumerates equivalent expressions of the generated equations, and then modifies the training objective of DSR. It is proved the equivalence of objective functions of DSR-Rex and DSR and reduced variance of the estimated objective using DSR. The performance of DSR-Rex is evaluated on Feymann datasets.

**Strengths:**

1. Improving the performance of symbolic regression is an important problem, and this paper is likely to have impact for the important method of DSR.

2. The motivating point of addressing equivalent symbolic equations is interesting and insightful.

3. The paper is clearly structured.

**Weaknesses:**

1. The presentation clarity of the paper needs to be improved, including notations and method details. Please see my detailed questions below.

2. The motivation needs to be strengthened to justify why numerically equivalent but symbolically different equations will pose challenges to DSR training and why the proposed methods.

3. The experiments can be enhanced by adding more benchmark comparisons.

**Questions:**

1. Details/notations about problem setup need to be more precise. For instance,
- for $\tau=(\tau_1,\dots,\tau_k)$, what is $k$? How is it determined?
- What is each $\tau_i$ -- is each of them a math operator/variable/coefficient?
- In equations (1) and (2), the reward is defined for each sequence $\tau$, but right after that, the notations override previous ones, where $\{\tau_1,\dots,\tau_N\}$ represents multiple sequences, so here each $\tau_i$ is a sequence, instead of an element in a sequence?

Please revise the notations and be rigorous about their meanings.

2. I understand that numerically equivalent but symbolically different expressions exist, and it is reasonable to try to avoid them. However, for the motivation of this work, I was wondering how this might negatively affect DSR. Why does it make it less stable or less efficient, as the authors claim?

3. Line 196, the sentence "Since we cannot directly use the probability distribution qθ to sample a group of sequences with the
same reward. Instead,..." seems to be grammatically incorrect.

4. More details of the method for equivalent expressions are needed for clarity: It is claimed that "In practice, equation 7 is not computed by enumerating every expression in Φ (as indicated by the inner summation)." and the details are in Section 3.2. However, Section 3.2 seems difficult to understand. What is the generated equivalent expressions for? How are they used in equation 7? Or is there an equivalent way to compute equation 7 after generating the equivalent expressions?

5. What if the equivalent expressions in Section 3.2 cannot enumerate all possible choices? What is the consequence, and how would limiting the number of them impact the results?

6. Setup for Section 5.2: Is Figure 2 the result for one dataset, or aggregating results from multiple datasets? Please consider showing the results for all 10 datasets.

7. How are the 10 tasks selected from the Feymann dataset? It would also be helpful to consider larger benchmarks like SRBench [1].

[1] La Cava, William, et al. "Contemporary symbolic regression methods and their relative performance." Advances in neural information processing systems 2021.DB1 (2021): 1.

8. The high-level idea of addressing numerically equivalent expressions seems widely applicable. Would similar ideas be useful beyond the context of DSR? It would be helpful to have some discussion on the broader scope.

---

> ### Author Response · Authors · 2024-11-26
> **Greatly appreciate for your constructive feedback!**
>
> ### 1. Notation Confusion
> Thank you for pointing out the notation inconsistency. In our work, $\tau$ represents a sequence of grammar rules, where each $\tau_i$ corresponds to one mathematical operator, variable, or coefficient. The maximum length of $\tau$ is set to be $k$. We previously used $\tau_1, \ldots, \tau_N$ to denote a batch of sequences, which may have caused confusion. We will revise the notation system throughout the paper to ensure consistency and clarity.
>
> ------
>
> ### 2. Impact of the Proposed Module on Classic DSR
> Classic DSR relies solely on the reward of an equation to guide its search. Over many iterations, it may implicitly learn numerically equivalent but symbolically different expressions. Our method explicitly incorporates this information into the model, encouraging it to adapt more quickly to such equivalencies.
>
> ------
>
> ### 3. Grammar Errors
> We will carefully proofread the content to eliminate grammar errors and typos, ensuring the manuscript is polished and professional.
>
>
> ------
>
> ### 4. Misinformation Between Equation 7 and Section 3.2
> Equation 7 provides the theoretical foundation for the proposed idea, while Section 3.2 describes its empirical implementation. The empirical results serve as an approximation of the theoretical estimator because, in theory, the group size can be infinitely large. Limiting the maximum group size introduces an approximation error. We will conduct additional ablation studies to analyze the impact of this approximation in detail.
>
> ------
>
> ### 5. Experimental Result Presentation
> We will rewrite Section 5.2 to provide a more detailed comparison with baselines, covering all instances in the dataset for a comprehensive evaluation.
>
> ------
>
> ### 6. Ten Tasks from the Feynman Dataset
> Thank you for raising this concern. Our intention was to demonstrate the effectiveness of the proposed method on challenging instances from the Feynman dataset, as summarized in Table 2. We will include a detailed learning comparison over these selected hard instances in a future revision.
>
> ### 7. Future extension to other base methods
> Thanks for your suggestion We will try to incorporate this idea into a wider range of baselines to show the effectiveness of the proposed idea.

---

### Official Review · Reviewer_ooAT · 2024-11-03

**Soundness:** 2
**Presentation:** 3
**Contribution:** 2
**Rating:** 3
**Confidence:** 4

**Summary:**

This paper presents Deep Symbolic Regression via Reasoning Equivalent Expressions (DSR-REX), an enhancement to deep reinforcement learning-based symbolic regression (DSR). The key innovation is leveraging numerically equivalent mathematical expressions to reduce policy gradient estimate variance while maintaining unbiasedness. The method incorporates a symbolic reasoning module that generates equivalent expressions through mathematical transformations, leading to improved convergence and performance compared to baseline deep RL methods. The authors provide theoretical guarantees for their approach and demonstrate empirical improvements on several datasets.

**Strengths:**

* Interesting approach to leveraging equivalent expressions for variance reduction in symbolic regression, with supporting theoretical analysis
* The methods to reason and find equivalent expressions are straightforward and fast, making them easy-to-use for future works.

**Weaknesses:**

* Limited evaluation scope using primarily trigonometric datasets and a small subset of Feynman equations, rather than standard benchmarks like SRBench (all Feynman equation, black-box datasets)
* Comparison against outdated baselines (DSR, neural guided GP) rather than current SOTA methods like PySR, uDSR, E2E, TPSR, and SPL
* Insufficient analysis of how the theoretical guarantees translate to practical scenarios, particularly regarding the sampling distribution of equivalent expressions
* Lack of ablation studies on the impact of different group sizes and reasoning rules

**Questions:**

1. The theoretical guarantees assume fair sampling of all equivalent sequences for each expression, but in practice this may not hold. Consider two expressions φ₁ and φ₂, where φ₁ finds only two equivalent forms, while φ₂ finds N>>2 equivalent forms through the designed reasoning rules. This could lead to q(φ₂) > q(φ₁) simply due to having more discoverable equivalent forms (e.g., there are lots of trigonometric equivalences compared to other operations), rather than actual learning preference. How does this potential bias affect the training process?

2. What is the value of max group size parameter, and how sensitive is the method to this parameter?

3. Could you clarify if the results shown in Fig. 2 (right) are averaged across all benchmark datasets or specific ones?

4. How are the 10 Feynman datasets selected? Why not evaluate on standard benchmarks like SRBench and compare against more recent SOTA methods?

---

> ### Author Response · Authors · 2024-11-26
> **Thank you for your valuable feedback.**
>
> ### 1. Limited Evaluation Scope
> Thank you for your feedback on the experimental evaluation. We appreciate your suggestion and will include the mentioned datasets in a future revision to broaden the evaluation scope.
>
> ------
> ### 2. Comparison with Recent Baselines
> Thank you for bringing recent methods to our attention. The proposed module is currently applied to the classic DSR method. We believe it can also be applied to and compared against recent methods in symbolic regression, and we plan to include such comparisons in future work.
>
> ------
> ### 3. Insufficient Theoretical Analysis
> We have included a detailed theoretical analysis in Appendix B, focusing on the improvement of the empirical variance of the policy gradient estimator. Could you provide additional details or suggestions on what further theoretical analysis could help demonstrate the effectiveness of the proposed module?
> 4. Impact of Hyperparameters on the Proposed Module
> Thank you for raising this concern. We will perform an ablation study to evaluate the impact of group size on the estimated policy gradient value and include this analysis in a future revision.
>
> ------
> ### 5. Potential Bias with Different Group Sizes
> We appreciate your comments on the effect of different group sizes. In Theorem 1, we demonstrate that the new objective (over probability q) is equivalent to the original objective (over probability p), indicating that group size does not theoretically affect the model. However, empirically, using a maximum group size to sample equations (rather than considering all equations in the group) introduces estimation bias and variance. We will conduct additional ablation studies to analyze this effect in detail in future work.
>
> ------
> ### 6. Experimental Analysis
> Thank you for your concerns regarding the experimental analysis. We acknowledge that the current experimental setting and results could be presented more clearly. In the future, we will carefully revise the content to ensure clarity and provide a thoroughly proofread version.

---

> > ### Comment · Reviewer_ooAT · 2024-11-28
> > **Response to authors**
> >
> > Thank you for your response. The proposed improvements could strengthen future revisions of this work. To address the theoretical-practical gap, I would encourage analyzing how the group size parameter and distribution of equivalent expressions affect performance through ablation studies. Additionally, comprehensive experiments on standard benchmarks could better demonstrate the empirical benefits of leveraging equivalent expressions.
> >
> > Given that these important changes remain to be implemented, I maintain my score.

---

### Official Review · Reviewer_zjYU · 2024-11-04

**Soundness:** 2
**Presentation:** 3
**Contribution:** 2
**Rating:** 5
**Confidence:** 3

**Summary:**

The paper identifies a problem of deep symbolic regression (DSR) for symbolic regression problems, that failure to capture equivalent expressions results in high variance of gradients and unstable training for the policy gradient estimator. The author proposed to address the problem by appending the symbolic reasoning module to a batch sampling of DSR to capture the equivalent expressions and adopting a new policy gradient method based on the group of equivalent expressions.

**Strengths:**

**1)** The paper is well written with clear notations, concrete technical details, and illustrative figures to explain the problem.

**2)** The paper is well-motivated by an interesting topic of expression equivalency in the symbolic regression (SR) area, which is promising to attain better performance with existing SR models and develop new SR models.

**3)** Theoretical analysis provides the performance lower-bound as DSR.

**Weaknesses:**

**1)** Expression equivalency problems exist in nearly all SR methods. Compared with the large landscape of SR model families, the baseline model DSR is a little bit out-of-date. For example, GPMeld, the successor of DSR in Figure 2, exhibits better performance than DSR, and a similar performance to DSR-REX.  Besides, the benchmarking models adopted in the experiments only encompass Reinforcement Learning (RL) based methods and one RL and genetic programming hybrid method GPMeld. To make stronger conclusions, more types of SR models should be considered, such as AI Feynman 2.0 as cited in the paper which studies similar expression equivalency problems.

**2)** Figure 3 only compares the efficiency between the steps within the DSR-REX with different architectures. The comparison of efficiency between DSR and DSR-REX would bring in more insights.

**Questions:**

**1)** Can you include more types of SR models in benchmarking, or explain the advantages of DSR-Rex over AI Feynman 2.0 in capturing equivalent expressions?

**2)** In equation (4), You mentioned that $\mathbb{I}$ \{ $\cdot$ } $=1$ if $\tau$ can be converted to $\phi$, however, according to the definition in line 85, $\phi$ is one specific expression. How do you obtain the probability of the equivalent group? Do you mean $\phi$ represents all equivalent expressions to $\phi$ here? In line 181, "all possible sequences" refers to all the sequences in the same equivalent group, or all the expressions have been sampled?

---

> ### Author Response · Authors · 2024-11-26
> **Thanks for your constructive feedback!**
>
> ### 1. Limited Evaluation Dataset and Baselines
> Thank you for your feedback regarding the experimental evaluation. We appreciate your suggestion and will incorporate the mentioned datasets and recent baselines in a future revision to broaden the evaluation scope. We believe the proposed module is well-suited for application to and comparison with state-of-the-art methods in symbolic regression. Such comparisons will be included in our future work.
>
> Thanks for the suggestions. We will extend the analysis in Figure 3 by experimenting with different neural network architectures.
>
> ------
>
> ### 2. Notation Confusion
> Thank you for highlighting the issues with our notation. We acknowledge the inconsistencies in both the notation and its definitions and will address these in a future revision. To clarify, with some slight abuse of notation, we use $\phi$ to denote a group of sequences that can be transformed into the equation $\phi$ (i.e., sequences that yield the same reward value). In line 181, the term "all possible sequences" refers to the entire search space of expressions.

---

> > ### Comment · Reviewer_zjYU · 2024-12-02
> >
> > Thank you for your clarification. To strengthen the quality of the experimental results, I recommend the author benchmark their method on the SRBench dataset with state-of-the-art baselines. Based on the current manuscript, I will keep my score.

---

### Note · Authors · 2025-01-17

I have read and agree with the venue's withdrawal policy on behalf of myself and my co-authors.